# Implementing Autonomous Driving Behaviors Using a Message Driven Petri Net Framework

**DOI:** 10.3390/s20020449

**Published:** 2020-01-13

**Authors:** Joaquín López, Pablo Sánchez-Vilariño, Rafael Sanz, Enrique Paz

**Affiliations:** Department Ingeniería de Sistemas y Automática, University of Vigo, 36200 Vigo, Spain; pasanchez@uvigo.es (P.S.-V.); epaz@uvigo.es (E.P.)

**Keywords:** autonomous driving, decision-making system, autonomous driving behaviors, vehicle control framework, Petri nets

## Abstract

Most autonomous car control frameworks are based on a middleware layer with several independent modules that are connected by an inter-process communication mechanism. These modules implement basic actions and report events about their state by subscribing and publishing messages. Here, we propose an executive module that coordinates the activity of these modules. This executive module uses hierarchical interpreted binary Petri nets (PNs) to define the behavior expected from the car in different scenarios according to the traffic rules. The module commands actions by sending messages to other modules and evolves its internal state according to the events (messages) received. A programming environment named RoboGraph (RG) is introduced with this architecture. RG includes a graphical interface that allows the edition, execution, tracing, and maintenance of the PNs. For the execution, a dispatcher loads these PNs and executes the different behaviors. The RG monitor that shows the state of all the running nets has proven to be very useful for debugging and tracing purposes. The whole system has been applied to an autonomous car designed for elderly or disabled people.

## 1. Introduction

Autonomous vehicles are one of the greatest engineering challenges of our era. Since the first successful demonstrations in the late 80s [1,2], great progress has been made in this field. However, achieving the highest level of autonomy (level 5 in the J3016 SAE document [3]) involves overcoming many technical, social, and legal challenges [4].

Driving decisions made by the vehicle control framework are based on the traffic situation and prior knowledge of the system. The traffic rules, street maps, and vehicle dynamics are among the prior knowledge of the system. The traffic situation is obtained by the perception system that includes observations from different on-board sensors (camera, sound navigation ranging (SONAR), Light Detection and Ranging (LIDAR), Global Positioning System (GPS), and odometry). The decisions can take the form of variables that control the vehicle’s motion.

These decision-making systems are usually implemented within four hierarchical components [5]:Route planning: Computes the route through the road map to reach the goal.Executive layer: Decides the high-level actions to follow the planned route according to the traffic rules. Since the driving process is a complex task that has to cope with many possible situations (stop, crosswalk, crossroad, overtaking, etc.), it is usually defined as a set of behaviors.Motion planning: The actions commanded by the behaviors must be converted, in the end, into a trajectory that must serve as a reference to the vehicle control loop. The trajectory can be defined as a sequence of points or a sequence of motion controls, depending on the planner used.Vehicle control: This controller selects the appropriate actuator inputs to carry out the planned trajectory. This module has to deal with the deviation errors when avoiding obstacles on the road.

The scope of this research takes in the executive layer and, in particular, the behavioral decision-making.

These systems are complex, so to support software development, the control framework will provide integration of loosely-coupled application software components to minimize covert interference [6]. For example, the Robot Operating System (ROS) is one of the most popular open-source platforms providing a dynamic middleware with publisher/subscriber communication and a remote-procedure-call mechanism. The different modules in the control architecture implement the aforementioned functionalities. These modules implement basic actions and report events about their state by subscribing and publishing messages.

In this paper, we propose an implementation of the executive layer based on Petri nets (PNs) using the RoboGraph (RG) tool [7]. PNs are a powerful tool to model, design, and analyze distributed, sequential, and concurrent systems [8]. RG uses the middleware inter-process communication mechanisms to coordinate the actions in the rest of the modules to execute a behavior.

Using PNs provides several advantages when defining the traffic rules and building the executive layer in the autonomous car application. They include: *Flexibility*, in which the traffic rules and behaviors are defined as PNs using the RG graphical user interface (GUI) [7], and they can be modified without the need to change the code; *Module abstraction*, which involves separating the modules interface with the PN from the implementation; *Reduced development time*, in which behaviors are implemented as PNs; *Easy maintenance*, which means tracing and debugging problems is easier when the system state can be seen by looking at the evolution of a PN rather than monitoring a set of variables; and *Analysis and test*, in which PN properties also make them good candidates for qualitative (un-timed models) performance evaluation and quantitative (timed models) performance evaluation of the car behaviors. Significant research has been carried out in this area for industrial applications [8,9] and also some in mobile robot tasks [10].

The main contribution of this research is to provide a new approach to implement autonomous driving behaviors using a discrete event model framework. Unlike previous solutions, this approach integrates the model definition with the model verification, execution and monitoring using a framework based on PNs. The advantages of using this kind of framework include:Behavior’s definition: A method to define the driving behaviors using message driving PNs. Traffic rules and behaviors are defined as PNs in a very intuitive and flexible way.Formal verification: Using this PN model representation, the formal verification can be included as part of the normal software development cycle as described in [11].Monitoring: PNs include a graphical representation with tokens that model the state of the system. The inter-process communication messages can be used to update the tokens and to display the state of the system in real time.Debugging: Many autonomous car software architectures, such as ROS, provide some way to store the inter-process communication messages. Those messages can be used later to represent the system state progress through the PNs.

The executive layer described here was implemented as part of the vehicle navigation system and applied to an autonomous car designed for elderly or disabled people described in [12].

The rest of this paper is organized as follows. The next section introduces the work related to this research. Section 3 presents the control architecture used in our mobile robots. After an outline of the *RoboGraph* tool in Section 4, the description of the executive layer is summarized in Section 5. Section 6 shows some examples of behaviors with the corresponding PNs. Results are presented in Section 7. Section 8 concludes the paper.

## 2. Related Work

The behavioral decision-making system is the executive control layer that manages the sequence of actions and events that take place in each possible driving situation. The different driving situations or scenarios include overtaking, yield, stop, crosswalk, crossroad, and traffic lights. The decision-making module should include tools to define the traffic rules for each one of these scenarios. Furthermore, they should also provide a way to define the sequence of actions to execute in response to all possible events that can occur in each scenario. Different approaches have been proposed to implement the decision-making system. Some heuristic solutions [13] are based on the concept of identifying the set of driving scenarios. To identify each one of these contexts, the vehicle will focus on a reduced set of environmental features.

One of the main challenges when implementing a decision-making system for autonomous driving is the complex interaction between the car and the environment. Even in the case of modeling all the possible states and situations, the uncertainty introduced by the perception system [14] will include uncertainty to assess the current state of the vehicle and scenario. Furthermore, the presence of nearby external agents, such as other vehicles [15] or pedestrians [16], makes it especially challenging because their behavior is, in general, unpredictable.

Decision-making for autonomous driving is challenging due to uncertainty in the knowledge about the state of the vehicle and particularly the driving situation. This uncertainty comes from different sources, such as that introduced by the perception system [14] and the observers. Especially challenging is to estimate the continuous state of nearby external agents, such as other vehicles [15] or pedestrians [16], since their behavior is, in general, unpredictable.

Research to improve decision performance can be focused on reduction of the uncertainty introduced by the observers or towards construction of a system that takes optimal decisions, assuming the noise information provided by the observers.

Improving the observers means providing accurate sensors and better detection and tracking methods. Much research has aimed to anticipate future intentions of other traffic agents [17]. The solutions proposed range from deterministic models [18] to different probabilistic models [19], such as Kalman Filters (KF) [20], Dynamic Bayesian Networks (DBN), and Hierarchical Dynamic Bayesian Networks (HDBN) [21] or Gaussian Process (GP) regression [22]. Most statistic observers provide observation with a probability value or a probability distribution. Modeling these distribution probabilities is not an easy task and many of the models mentioned use some kind of reinforcement to learn some parameters of the estimated model or even the driving styles of other vehicles.

To design an optimal decision system that takes into account uncertainty, Partially Observable Markov Decision Processes (POMDP) offer a theoretically grounded framework to capture the fact that the decision system does not know the underlying state of the system. Instead, the POMDP maintains a probability distribution over all of the states and makes the decision according to this distribution. The probability distribution is updated with the observations and the actions carried out according to the transition and observation probabilities that define the system model. Even though, in theory, the solution of a POMDP provides the optimal solution for the model given the previous observations and actions, solvers [23] often have difficulty scaling computationally to real-world scenarios. Still, some approaches, such as that in [15], consider only a finite set of a priori known policies to capture a different high-level behavior from the other traffic agents. POMDPs take into account both the uncertainty in observations and the uncertainty in the outcome of the actions. Building the POMDP model is not an easy task because it is necessary to define not only the states, actions, and transitions but also all the transition probabilities and rewards. Some of those parameters can be learned from experimental data using some reinforcement learning or deep learning techniques [24,25]. A question remaining is to evaluate the performance of systems of this kind against other solutions, such as simpler finite state machines, where it is easier to model the traffic rules and change the model as the traffic rules are updated.

In theory, POMDP is a good framework for including uncertainty in the behavioral decision-making system. In practice, real-world scenarios are very complex with a large number of states, observations, and actions. Unfortunately, these kinds of POMDP models cannot scale to complex real-world scenarios. Instead, other approaches deal with uncertainty in the observers using some Bayesian model to estimate the state of the system and a simpler discrete event system to model the behaviors. Some popular solutions are hierarchical finite state machines [26], finite state machines (FSM) [27], and decision trees.

The Defense Advanced Research Projects Agency (DARPA) Urban Challenge [28] contributed to the development of some of these technologies. The behavior generation component [29] for the Tartan team (winning team) was implemented using a hierarchical finite-state machine. The main task (mission) is hierarchically decomposed into different behaviors. The top-level hierarchy behaviors are lane-driving, intersection handling, and achieving a pose [13]. A similar hierarchical finite-state machine [27] was used by the Standford University team that finished in second place. Thirteen states define the top-level hierarchy behaviors. The top-level behaviors of the California Institute of Technology include: road region, zone region, off-road, intersection, U-turn, failed, and paused. Ohio State University [30] also chose an FSM for their car.

Even though FSMs can deal with relatively large problems, they need to explicitly define all states. In some situations, where it is necessary to define an action in small actions that can run in parallel or when a system can be divided in subsystems, the number of states grows exponentially. This is known as the state explosion problem, and that is one of the reasons why some researchers are considering Behaviors Trees (BT) instead [31]. BTs provide a lot of flexibility and can be used to implement the behavioral layer architecture in autonomous driving [32]. BTs are also easier to maintain and to expand [33].

Another option to overcome the FSM state explosion problem is to use PNs. They have been used for different purposes in autonomous vehicles: evaluating the mission reliability [34]; designing and modeling a cruise control system [35]; modeling an intersection collision warning system [36], and modeling the cooperation of a human driver and an automated system [37]. On the other hand, in the approach presented in this paper, PNs are used to model the desired behavior of the vehicle in different traffic situations.

Several properties of PNs, such as concurrency, lead to simpler models that can deal with a large number of states compared with FSM. All of these properties come from the fact that the state of the system in a PN is represented by a set of tokens associated with places, while in a classical FSM every place represents a state. Therefore, in a PN using only a few places, a large number of states can be represented through a different combination of tokens. The problem of state-space explosion in PNs comes with the reachability analysis because somehow all the states need to be unfolded. To solve that problem, we use hierarchical PNs, as described in [11]. The hierarchical interpreted PNs are used to model the behaviors for each driving situation or scenario. Additionally, a main PN defines the conditions to switch between the different behaviors. This main PN can start and stop other behaviors (PNs) whenever the driving situation changes. The PNs used here are binary, which means that the tokens associated with the places can only be one or zero. The hierarchy is used to break down the complexity of the model so that a PN can be started from another one. A full description of the reachability analysis of these hierarchical model is described in [11].

Subsumption architecture has been extensively used at different levels in the autonomous robots field. The architecture focuses on sensing and reaction, and it builds small systems that could cope with uncertainty. For example, the research presented in [38] provides an autonomous vehicle with the means to stay within the boundaries of a course, while avoiding obstacles. Developments in cognitive systems are also the base for some biologically-inspired layered control architecture for intelligent vehicles, such as in [39].

## 3. Software Architecture

The control architecture used for the navigation system (Figure 1) includes a set of independent modules running as independent Linux processes. These modules share information using the inter-process communication system provided by ROS. According to the functionality of these modules, they can be organized in four sets:The hardware driver includes all the processes that manage the hardware devices on board the vehicle (sensors and actuators).The control layer implements the basic navigation functions: reactive control or obstacle avoidance (*local navigator*), localization (*localization*), path planning (*map manager*), and perception (*event monitor*) that process information from sensors to detect the basic events [40].The executive layer coordinates the sequence of actions that need to be executed by other modules to carry out the current behavior. This layer is the goal of this research.The interface layer consists of a set of processes to interact with the users and the web.

Some navigation modules included in the reactive control are critical for the safety of the application. These important modules are mainly located in the lower layers: in the hardware layer, the actuators that control the vehicle and the sensors that detect the obstacles; in the hardware driver layer, the drivers that control those devices; and in the control layer, the local navigator. Unlike other architectures, such as in [13,41], which includes planning in the motion control, here, it is divided into high-level planning and lower-level reactive control. High-level planning calculates a path consisting of a sequence of lanelets [42]. The goal of the local navigation system is to follow the path safely, keeping the car within the driving lane.

### 3.1. Interface Modules

There is a main GUI (user GUI) that shows the map, the planned trajectory and the position of the vehicle in the map. This GUI can be accessed onboard or remotely. Users can also use other onboard GUIs to debug the system, test hardware devices, show sensor readings, and manually move the car.

### 3.2. Executive Modules

RG was first developed by our group [7] to work with IPC (inter-process communication [43]) and JIPC (java inter-process communication [44]). In this research, was been extended to work within ROS. RG includes two independent modules: a GUI to edit, monitor, and debug the tasks (PNs), along with a dispatch that executes the tasks. A detailed description of this module will be presented in Section 5.

### 3.3. Control Modules

The control modules provide the basic navigation functions:Path planning. The *map manager* module carries out all the map related functions. The map is based on lanelets. Each lane is delimited by two ways [42]. This module provides the path as a sequence of lanes. It is also in charge of providing all the information related to the map. For example, at some point, to overtake other car, the *map manager* module could be asked about contiguous lanes. The maps also include the position of the traffic signs and regulatory elements that need to be provided to the executive layer.Obstacle avoidance. The *local navigator* module implements the method described in [45] to follow the path in a safe way. This algorithm is a modified version of the BCM (beam curvature method) reactive control method [46] and is able to avoid obstacles that partially block the lane. If the obstacle blocks the lane completely, the car has to stop.Localization. The *localization* module is a Kalman filter that integrates information from the GPS, encoders, and map features.

## 4. RoboGraph

Figure 1 shows the two main programs that form *RoboGraph*. The GUI is a development tool that makes creating, editing, and monitoring of the different tasks possible, while the DISPATCH is in charge of executing those tasks.

### 4.1. GUI

This program is used to edit the PNs, monitor the execution of the PNs, and playback a finished execution of the system.

It includes a simple and intuitive PN graphical editor to create new tasks as a sequence of places and transitions. The PN is constructed by selecting and dragging different elements (places, transitions, arcs, and tokens). Afterwards, actions associated with places and transitions, as well as conditions associated with transitions, can be defined for the interpreted PN.

The actions associated with places and transitions can be commands implemented in any module in the control architecture shown in Figure 1. The commands can be selected from a list automatically generated by the GUI. In a similar way, events can be part of the firing conditions of a transition. Some events are related to the arrival of a message from another module. The condition to fire the transition may only be the arrival of the message or a condition on some parameters of the message. For example, a condition can be that the event manager publishes a *STOP signal detected* message or that the parameter distance of the *STOP signal detected* message is less than ten meters. Figure 2 shows an example in which a transition is selected and the list of possible commands and events that can be associated with this transition is displayed. In Figure 2, light pawns represent command messages that are executed when the transition is fired, and dark pawns represent events that integrate the condition associated with the transition.

There are other actions that can be associated with places and transitions besides commands, such as timers. A timer can be started in a place or transition for a limited period of time. The timeout can be part of the firing conditions of a transition.

Global variables are used to get starting data and store information to share conditions and events in different places and/or transitions. A condition to evolve the system might depend on the data coming from two or more messages.

Once the behavior is defined as a PN, the GUI includes an analyzer to test different problems and properties, such as possible deadlocks.

During the execution of the system, many independent modules (Figure 1) are running in parallel. Debugging in these conditions is quite difficult. When the system crashes or does not work correctly, it is hard to identify the problems. The first step is to identify the module or modules that cause the problem. Sometimes the problem is not in a particular mode but in the synchronization between modes. The RG GUI can be very helpful for this task. While the system is running, the GUI in monitor mode connects to ROS and subscribes to different dispatch messages that show the status of the different running or waiting PNs. These messages are published by RG dispatch every time a PN starts, finishes, or evolves. The GUI shows every running PN in a different tab with the current marking. This way the status of the behavior can be identified in a snapshot, looking at the tokens in the PN. If the PN gets stuck in a particular marking, the problem can be easily identified by looking at the conditions of the enabled transitions. The typical situation is that a condition never holds because a module is not working the way it is supposed to work or the condition is not well defined. In either case, the module responsible for the problem can be easily identified.

When the autonomous car is operating in the real world, the number of PNs running in parallel and processes included in the system is huge. In this situation, debugging in real time becomes a hard task since many PNs can be evolving at the same time. In addition, some problems are hard to reproduce and it is useful to be able to playback a failed execution. In order to be able to deal with these problems, an XML log file with Dispatch ROS messages is created at runtime. Researchers can then run the GUI in playback mode, open the log file, play it at the same pace as in the real execution, and stop it at any time. It can also be traced step-by-step or jump to a defined place in the execution and look at the messages that arrive to RG.

### 4.2. Dispatch

Dispatch is the RG module that loads the PNs and executes them. In some way, dispatch coordinates the actions of the vehicle control modules (Figure 1) according to the model defined in the PNs. The way RG dispatch interacts with the other modules in the control architecture is through messages. While executing a PN, the ROS messages associated with places are published every time the place gets a token (mark). A similar action takes place for the command ROS messages associated with transitions when the transition is fired. On the other hand, when RG starts the execution of a PN, it subscribes to all the the messages associated with transitions as events. Every time RG receives one of these messages, the conditions associated with the enabled transitions associated with the event are evaluated. Firing a transition results in execution of the commands associated with the transition, removing tokens from input places, and adding tokens to output places, according to the PN theory [47]. It is important to notice that only the publish/subscribe paradigm is used to provide non-blocking communications. This way, even if some other module crashes, dispatch will not be blocked.

Dispatch is subscribed to the PN execution or cancellation ROS messages. Every requested message has an owner, priority, and execution mode (serial or parallel). Priority and owner define the execution precedence and the ability to stop or kill a PN. When a PN execution message arrives, dispatch loads the corresponding XML file and links the associated objects. Then, it starts the execution of the PN by setting the initial marking and subscribing to all messages associated with transition conditions.

For debugging purposes, this module publishes ROS messages reporting the state of all running PNs. Each change in the status of a PN (start, stop, evolve) is reported by issuing a new ROS message that is published and stored in a log file.

The RG tool was developed in Java because of its cross platform and its dynamic class loading features. The structure of each PN is stored in an XML file. The code associated with each place and transition is stored in a Java class that is dynamically loaded when the execution of a new PN is requested.

## 5. The Executive Layer

The executive layer implements different behaviors needed to drive the car autonomously. In the context of this work, a behavior corresponds to a traffic situation, such as a traffic light, a pedestrian crossing, an intersection with a yield, an intersection with a STOP, entering a highway, etc. Each of these behaviors is implemented in one or more PNs. In some way, the PNs that implement a behavior define the traffic rules to handle the corresponding traffic situation. The resulting system is very flexible since a modification in the traffic rules is easily implemented with minor changes in the corresponding PNs.

The role of the executive layer is to coordinate and synchronize other modules to carry out autonomous driving in different situations. Every behavior is modeled as a sequence of discrete events using the PNs. While running, other modules communicate events to this layer (pedestrian crossing, intersection ahead, etc.) through messages. According to the PN receptivity, in every situation, i.e., behavior, the executive module is subscribed to the messages that can evolve the state of the PN. At the same time, the executive module can request the execution of some commands (reduce speed, look for pedestrians, etc.) publishing messages to other modules that can carry out these commands.

### 5.1. Starting the System

There is a default PN running in the background all the time. This net is waiting for a message from the user requesting to execute some of the tasks the car can carry out. Figure 3 is a simplified version of that PN.

The foreground color of places and transitions (dark or clear) is used to represent different information about the execution of the PN. For example, transitions subscribed to a message are represented with dark foreground if the modules that publish those messages are already running in the system. In a similar way, places that are associated with messages will be represented with dark foreground when some module has already been registered as publisher of that message. For timed transitions, the black foreground represents timers that were used in the last execution.

When the *RG dispatch* module loads a PN, it subscribes to all the messages that take part in the events of the PN and registers publishers for all the messages that are included in the actions. All the PNs presented here have the same starting sequence: a first place (with the initial marking) and a transition. In the place, a timeout of 300 ms is started. The end of the timeout is the firing condition of the transition. This timeout has to be included to deal with ROS delays to avoid the use of messages before being registered. To simplify the figures, these first two items will be omitted in the rest of the paper.

As soon as the timeout concludes, the timeout transition will be fired, the token from the *ini* place will be removed and a new token will be added to *wait* place. In this situation, transitions *manualDrive* and *goToPoint* will be enabled. One of them can be fired whenever the condition associated with it shows as true. The condition associated with transition *goToPoint* is the reception of a message from the user GUI module selecting a destination. Upon reception of this message, the token from the *wait* place is removed and a new token is set in the *wait start* place. After the user selects the destination and a path is generated, the start button can be used to command the car to follow the path. In this case, the start message is published by the GUI and received by dispatch that fires the *start* transition. When the transition is fired, a token from *wait start* is removed and a new token is added to the *RUN_start* place.

The action associated with the *RUN_start* place is to send a message requesting to run the PN that starts one of the different behaviors (*selector* PN). The rest of the PN (left branch) deals with the selection between manual and autonomous drive. A transition with the STOP button event has been omitted for the sake of simplicity.

### 5.2. Monitoring the Behaviors

The *selector* PN decides which behavior to run, according to the traffic situation, and monitors the execution of the behaviors.

Figure 4 and Figure 5 represent the main parts of the *selector* PN. The dangling arc coming from the right of Figure 4 is connected to the dangling arc leaving from the left of Figure 5. The other dangling arc of Figure 5 is used to connect another part of the PN used for debugging purposes, but it has been omitted here for the sake of simplicity. Figure 4 represents the part that decides what behavior to execute and stores the name of the behavior in a global variable (*mainBhvr*). A place labeled as *wait* gets a token as soon as the PN is started and waits for the events that are relevant to make the decision. Whenever an event that is likely to change the traffic situation occurs, it should be detected by the corresponding module (*event monitors*, *map manager*, etc.) that will send the corresponding message. That message should trigger the related *wait* output transition. Therefore, the token from the *wait* place is removed and a new token is added to the *calculateNewBehaviour* place. In this place, the main behavior is updated and stored in a PN global variable (*mainBhvr*). It is important to note that standard PNs do not include variables, but they are an added feature in RG [7] that simplifies the net in applications of this kind.

The *mainBhvr* variable is used in the other part of the *selector* PN (Figure 5) to decide when to start or stop a behavior. As in the previous case, the *selectBH* place gets a token as soon as the PN starts. When the *mainBhvr* variable is set to one of the behaviors that matches the value associated with one of the *selectBH* output transitions, that transition is fired. Therefore, the token from the *selectBH* place will be removed, and a new token is added to one of the behavior places. The action associated with most of the behavior places is to start the corresponding PN. As soon as the traffic situation changes, the running PN that implements the current behavior should finish. Places labeled as *wait_END_OK* in Figure 5 should get a token waiting for the message that is sent when the behavior PN finishes. If that message arrives before the timeout, the *END_PN* transition will be fired and finally the *selectBH* place will get a token, meaning that the system is ready to run the next behavior. If the timeout occurs while the *wait_END_PN* place is marked with a token because the *END_PN* message did not arrive, the *timeout* transition will be fired and a message to kill the behavior PN will be issued, forcing the behavior to finish and popping up an error message for the user.

It is important to note that safe navigation is guaranteed at all times by the control loop on the *local navigation* module. Any kind of obstacle should be detected by the range lasers and avoided by the *local navigation* module, using a modified version of the CVM (Curvature Velocity Method) obstacle avoidance algorithm named CVM-Car [45].

In the next section, we will describe some of the behaviors implemented in the autonomous driving system.

## 6. Behaviors

### 6.1. Pedestrian Crossing Behavior

A pedestrian crossing behavior is started when the vehicle approaches an area where the corresponding regulatory signs are displayed. This behavior is defined by the PN of Figure 6.

After the starting sequence, the place *watchForPedestrians* gets a token, and the associated actions are executed. In this case, the actions consist of sending a message to the *event monitor* module to watch for pedestrians crossing on the crosswalk. There are three possible outcomes of this state regarding the event produced by the *event monitor* module: a message informing that so far no pedestrian has been detected (*notPedestrian* transition); a message reporting that a pedestrian has been detected (*pedestrian* transition); and no confirmation message has been received from the *event monitor* (timeout transition).

If no pedestrian is detected, transition *notPedestrian* is fired, removing the token from *watchForPedestrians* and adding a token to the *followPath* place. While in this place, the *local navigation* module will keep following the lane provided by the *map manager* module. The system in this state is receptive to two events: *pedestrian* and *close2Crosswalk*.

The first event (*pedestrian*) is a message published by the *event monitor* module; if this message is received, the *pedestrian* transition will be fired. According to the transition firing rules, a token is removed from the *followPath* place, and a token is added to the first *stop* place.

The event *close2Crosswalk* occurs when the robot is closer than a threshold distance (D) to the crosswalk. The condition associated with transition *close2Crosswalk* compares the current robot position with the crosswalk position. The vehicle position message is published by the *localization* module (Figure 1). The *RG Dispatch* module that executes the PNs is subscribed to all messages used in the enabled transitions. Every time one of these messages arrives, *RG Dispatch* evaluates the enabled transitions and fires the ones evaluated to true. On this occasion, the distance between the vehicle position (vehicle position message) and the crosswalk is obtained and compared with the threshold D.

When the transition *close2Crosswalk* is fired, a token is added to the reduceVel transition (Figure 6). The principal action associated with this transition is to command the local navigation module to reduce the speed. Still, the vehicle keeps following the path until the crosswalk is reached (*crosswalkReached* transition) unless a pedestrian is detected (*pedestrian* transition). If a pedestrian is detected, the *pedestrian* transition will be fired upon the reception of the corresponding message published by the *event monitor* module. A token will be added to the *stop*, and a message commanding *local navigation* module to stop in front of the crosswalk will be published. After the pedestrians cross and no pedestrians are detected, transition *notPedestrian* is fired, and the vehicle starts moving again.

Finally, when the *END* place is reached, *RG Dispatch* will end the PN issuing an *END_PN* message that will be used by the *selector* PN (Section 5).

### 6.2. Intersection Behaviors

Intersections pose many challenges as vehicles need to deal with many different maneuvers, such as merge, left turn, yield, and stops at certain road intersections. They include also a considerable number of different elements: traffic signals, traffic lights, turning lanes, merging lanes, and other vehicles entering the intersection.

Figure 7 shows the PN that represents the sequence of events and actions defining the behavior of the car in a traffic STOP situation. After the starting sequence, the *stop* place will get a token. That will lead to publishing a couple of messages: the first one is addressed to the *event monitor* module to check for vehicles at the intersection, and the second one is addressed to the *local navigation* module to stop before the intersection. The firing condition associated with the *stopped* transition is the reception of a car’s velocity message from the *local navigation* module with velocity equal to zero. When this transition is fired, a timer of one second is started and the place labeled “*wait 1s for safety*” gets a token. When the timer finishes, the timeout transition is fired, and place labeled “*wait safe merge*” gets a token. This state is sensitive to two events associated with the corresponding transitions. The *notSafeMerge* transition is fired if a an unsafe merge message is received from the *event monitor*. The other event is that a safe merge message is received, in which case the *SafeMerge* transition would be fired. The state of the system can move between these two states until the car is in the *followPath* place long enough to enter the intersection. Once the car is in the intersection, it proceeds to cross the intersection since there is no point in stopping in the middle of the intersection. In any case, as aforementioned, safe navigation is guaranteed by the *local navigator* module.

The yield behavior is very similar to the STOP case, with the main difference that no stop is needed when it is safe to cross. That leads to a PN similar to the one in Figure 7, removing the second and third places and transitions.

The sequence for the traffic light situation is defined by the PN in Figure 8. After sending a message to the *event monitor* to check for the traffic light at the intersection (*checkTrafficLight* place), the car might need to stop (*stop* place) sending a *stop* message to the *local navigation* module, or continue (*followPa**t**h* place). The car will stop (*stop* place) if one of these three different sequences occurs:The traffic light is red (*red* transition).A timeout expired before getting any traffic light reading from the *event monitor* module (*timeout* transition).The traffic light is amber (*amber* transition), and it is safe to stop before the traffic light (*safeStop* transition).

The car will continue (*followPath* places) if one of these sequences occurs:The traffic light is green (*green* transition).The traffic light is amber (*amber* transition), but the car is already in the intersection, and it is not possible to stop in a safe place (*notSafeStop* transition).

There are other similar behaviors, such as entering a roundabout, entering a highway, etc., in which descriptions are not included here because the PNs that define their behaviors are very similar to some of those already depicted.

### 6.3. Overtaking Behavior

Overtaking is one of the most challenging maneuvers [48] that involves the cooperation of several modules. The *map manager* has the information to decide if in that part of the road the overtaking maneuver is allowed and for how long. The event monitor can decide whether it is safe to switch lanes according to the traffic. The vehicle needs to switch to the adjacent left lane and stay on that lane until the overtaken vehicle is left behind and there is a gap in the right line to return. Figure 9 represents the PN that models this maneuver. There are three main states: the first one is when the car is switching to the left lane (*switchLeftLane* place), the second one is while the car remains in the left lane (*overtaking* place), and the last one is when the car is switching back to the right lane (*returnRight* place).

While the behavior is in one of these states, it is sensitive to the messages that could change to a different state. For example, at the beginning (*start* place), the decision to start the overtaking maneuver (*evaluateCond* place) will depend on two messages reported by the *event monitor* module and one from the *map manager*. The *event monitor* reports when the adjacent left lane is clear for a sufficient distance (*leftLaneInfo* transition) and when there is a significant difference between the desired speed of the ego car and the speed of the car in front of it (*frontCar* transition). The *map manager* module knows the distance remaining in the current path where overtaking is allowed. Whenever one of these messages is received, the conditions are evaluated (*evaluateCond* place) and, depending on the outcome of this evaluation, the car might start switching to the left lane (*switchLeft* place) or keep waiting (*start* place).

Similar sequences are also defined while the car is returning to the right lane (*returnRight* place).

Both lane-change operations are depicted in Figure 10. When the local navigator receives a *switch2LeftLane* command together with the *overtaking lane* message, it changes the left lane border to the far side of the overtaking lane. In Figure 10a, the car is moving behind a slower car, and the executive module decides to overtake when it is legal to do so. The white circles on top of the lane borders are virtual obstacles generated by the CVM-Car obstacle avoidance method to keep the car in the lane. Figure 10b shows that the left way has been switched and virtual obstacles appear on the left side of this lane. Now, CVM-Car will try to avoid the other car by moving to the opening area, which is in the left lane. Once the car has overtaken (Figure 10c), it will return to the middle of the opening area, which is in the middle of the road. In order to return completely to the right lane, the left way progressively moves to the left way of the right lane.

## 7. Results

The executive layer described here has been implemented in a ROS node according to the architecture presented in Section 3.

The car shown in Figure 11 is based on an Open Motors Company electric TABBY EVO (Tabby Evo, 2018). The automation of the car was carried out by the ROBESAFE team of the University of Alcalá [49]. The local navigation module commands the translational velocity and the rotation angle of the front wheels and uses the odometry and the readings from a LIDAR Velodyne (VLP-16) located on top of the vehicle. More details about the car and the research carried out in other modules of Figure 1 can be seen in [50,51,52,53].

### 7.1. Simulation and Road Tests

Several tests have been carried out in two environments: the campus of the University of Alcalá de Henares and the closed circuit used in the International Conference on Intelligent Robots and Systems (IROS) 2018 autonomous vehicle demonstration event, both located in Madrid. A section of the Universidad de Alcalá de Henares (UAH) campus lanelet map is presented in Figure 12a, and the aerial view of the same section is shown in Figure 12b.

The first tests used a simulated version of the car and its environment. A model of the University of Alcalá environment was implemented in V-REP as described in [54]. The map of the simulated environment has been modeled using *OpenStreetMap*. The OSM environment file was converted to OBJ using *OSM2World* tool to be imported in V-REP, and then some elements were manually added for simulation purposes.

The main difference between the real and simulated cases is the performance of the monitors. What follows is the timeline of events and actions that take place in a couple of simulated scenarios and a final scenario with the physical system that shows the robustness of the framework regardless of some problems on the monitors.

#### 7.1.1. Intersection with Stop

Figure 13 shows an intersection with a stop scenario in the University of Alcalá campus. The ego car is turning left at the stop but has to give way to the oncoming cars in the left lane.

Figure 14 shows the sequence of events, evolution of the states in the PNs and velocity commanded to the car. It is important to notice that some of the states (markings) are active for a very short period and are hardly noticeable in Figure 14. The *start* PN is active while the system is working. Using the GUI, the driver can request the car to go to a point. That will trigger a pair of events (messages). The first one is *goToPoint.* When RG *dispatch* receives this message, it changes the marking of the *start* PN (Figure 3), removing the token from the *wait* place and adding a token to the *wait_star* place. When the route is planned, the *start* message is issued. On reception of this message, dispatch removes the token in the *wait_star* place and adds a token to the *RUN_start* place. There are two actions associated with the *RUN_start* place: first to publish a message (*run_PN*), requesting the execution of the *selector* PN (Figure 4 and Figure 5), and second, to start the default behavior (*follow_path*). The car starts in the middle of a straight section where it can reach the maximum speed (9 m/s) until it gets close to the place where it has to turn left, reducing its speed to 3 m/s, as shown in the bottom part of Figure 14.

The *mapManager* module publishes information about the next regulatory element in the road (stop, yield, crosswalk, etc.). The *selector* PN is subscribed to this message and when the next regulatory element (stop, in this case) is closer than a threshold distance (30 m), the PN marking changes, removing a token from the *selectBH* place and adding one to the *RUN_BHstop* place. The action associated with this last place is to publish a message to request the execution of the PN associated with the behavior (stop, in this case). This PN (Figure 7) commands the car to stop at the intersection. About 12 s later, the car reaches the STOP line and *localNav* stops the car and that event evolves the stop PN to a new state where it requests the *monitors* module to check the intersection. Since there is a car coming along the other lane, a *notSafeMerge* is issued by the *monitors* module and the car remains stopped for about 8 s. As soon as the intersection is clear, the *safeMerge* event fires the *safeMerge* transition evolving the *stop* PN to continue the path until the car reaches the middle of the intersection that finishes the stop behavior and continues the default *follow_path* behavior. Since there is a crosswalk at the beginning of the new lane (Figure 13), a similar process to the one described for the stop begins for the crosswalk behavior (Figure 14).

#### 7.1.2. Entering a Roundabout

In this scenario, the ego car is entering one of the roundabouts on the University of Alcalá campus as shown in Figure 15.

The sequence of events, evolution of the states in the PNs and velocity commanded to the car are shown in Figure 16. The sequence of events and actions for the *start* and *selector* PNs is the same as in the intersection with a stop described above.

The car starts in the middle of a straight section, where it can reach the maximum speed (9 m/s). Before reaching that speed, it gets close to the roundabout and, because of the curvature of the road, the local navigation module starts reducing the velocity, as shown in the bottom part of Figure 16. The maximum speed is reduced still more (to 3m/s) by the executive layer, according to the *crosswalk* PN that implements the crosswalk behavior (Figure 6).

The *mapManager* module publishes information about the next regulatory element in the road (stop, yield, crosswalk, etc). When the *selector* PN is notified by the *mapManager* module that the next regulatory element (crosswalk) is closer than a threshold distance (30 m), the PN marking changes, removing a token from the *selectBH* place and adding one to the *RUN_BHcrosswalk* place. One of the actions connected with this place is to publish a message to request the execution of the PN associated with the behavior (*crosswalk*). This PN (Figure 17) commands the car to proceed and reduce speed when it is close to the crosswalk, unless a person on the crossing is detected by the monitors. In this case, no pedestrian is detected (Figure 16) and the *crosswalk* PN is completed as soon as the car is on top of the crosswalk. The ending of the *crosswalk* PN is notified to the *selector* PN that evolves to the point where it is waiting for the conditions of another behavior to start.

In this scenario, the next behavior is the roundabout where it has to yield to other cars. The *yield* PN implements the yield behavior (Figure 17). Figure 16 shows the events that influence the sequence of this net. In this case, the *mapManager* module reports the proximity of the regulatory element (yield traffic sign) that causes the starting of the *yield* PN. A *safeMerge* event from the monitors causes it to position the token on the *followPath* place of the *yield* PN. When the car gets close to the traffic sign (*yieldClose* event produced by *mapManager*), the maximum velocity of the car is reduced again. Before the ego car gets to the roundabout, a couple of *notSafeMerge* events cause the car to stop (Figure 16), even though the monitors module did not detect the car for a few seconds, producing a *safeMerge* message in the middle. Finally, the car on the roundabout moves away and the ego car proceeds after reception of a *safeMerge* event from the *monitors* module.

#### 7.1.3. Crosswalk with Pedestrians

This scenario shows further advantages of the framework presented here, which has a clear division into high-level planning and lower-level reactive control when dealing with some problems caused by uncertainty in the observation system. When using real images from the car cameras, the monitors system is likely to provide some false positives or negatives. A pedestrian crosswalk was already included in the last simulation scenario for the case when no pedestrian crosses at the same time as the car. The scenario in this case is shown in Figure 18. It includes a couple of cars that partially block the path of the ego car and pedestrians crossing in front of the car.

Figure 19 shows the section of the UAH map (Figure 12a) where the crosswalk with pedestrian scenario takes place. This is a section of one of the road tests carried out in the UAH campus. The car starts from the right in autonomous mode with three people on board. For safety, if the driver uses the controls, the car switches to manual mode. As we pointed out before, the event monitor has to deal with several sources of uncertainty and can produce false events. However, it is important to notice that the behavior (PN) is only receptive to a few events in each instant. In order to save computing resources, the executive layer sends messages to enable and disable events according to the state of the PN.

The sequence of events, evolution of the states in the PNs and velocity commanded to the car are shown in Figure 20. The sequence of events and actions for the *start* and *selector* PNs is the same as in the scenarios described above.

The ego car starts 50 meters away from the first obstacle (red car, in Figure 18). Therefore, at the beginning it accelerates to the maximum speed in that section (9 m/s). Soon it encounters the first unexpected obstacle, and the reactive control needs to reduce the speed in the maneuver to avoid it. The velocity represented is the velocity commanded by the reactive control. Obviously, the car cannot follow this sudden change. The executive layer does not have any knowledge of the two cars on both sides of the road. The reactive control allows navigation without colliding with both obstacles, since there is still room in the lane to pass. Notice that the maximum velocities commanded by the executive layer are just references and the reactive control might not reach them, as in this case. At about second 12, a message from the *mapManager* module, regarding the proximity of a crosswalk, triggers the start of the *crosswalk* behavior. This behavior commands the vehicle to follow the path until a pedestrian is detected and reported by the monitors at second 14. The new state of the PN commands the reactive control to stop in front of the crosswalk. The reactive control reduces the car’s speed for two reasons: the first is because it has to stop soon and the second is to navigate to avoid the second obstacle. Between second 26.5 and 28.2, monitors report the presence and non-presence of pedestrians several times. Therefore, the stop at *crosswalk* and *recuceVel* commands are sent to the reactive control that does not need to change the car’s speed much, since it is already moving slowly. Unfortunately, due to a problem with nearby pedestrian detection, the monitors trigger a final no pedestrian message at second 18.18. That message, according to the crosswalk PN (Figure 6), commands the reactive control to proceed slowly. Luckily, despite the received command, the reactive control stops the car at second 19 because there is an obstacle and so it does not run over the pedestrian.

### 7.2. Efficiency Analysis

In order to evaluate the efficiency of the approach presented in this paper, real-time data were recorded on the occurrence of every event produced during several real road tests. The reaction times were recorded while the vehicle navigated around the University of Alcalá de Henares campus. The time distributions for these executions are displayed in Figure 21. Dispatch was executed in an Intel(R) Core(TM) i7-6700 CPU at 3.40GHz.

The solid line in Figure 21 represents the elapsed time in ms between one event arriving at the executive layer and the corresponding reaction command being issued. This includes the time to process the event, evolve the PN and issue the commands associated with the new state (PN marking). Reaction times for the executive layer follow a normal distribution with an average of 47 ms.

The dotted line represents the elapsed ms between the time a new PN execution is requested by the executive layer and the time the initial marking is set. As described in Section 4, this includes the time to load the XML file and Java classes associated with places and transitions, create the PN structure and set the initial marking.

Finally, the solid line represents the time delay estimation added by the ROS inter-process communication mechanism. That is, the elapsed time in ms from a module publishing a message to it being received by another module. Obviously, that includes the effect of the process looping time that is set to 40 ms.

## 8. Conclusions

By using a Petri-net-based programming environment to implement the executive layer of a car navigation framework, we have shown that it is possible to build and maintain the behavior system in a relatively fast and intuitive way. The programming environment, named *RoboGraph*, allows the edition, execution, tracing, and maintenance of the PNs that implement the behaviors needed for different traffic scenarios. RG has been used before within the RIDE framework [44] for several multirobot applications, such as on the hotel assistant BellBot [55], the surveillance system WatchBot [56], and a tour guide system GuideBot [57]. For the work presented in this paper, the RG inter-process communication mechanism has been extended to communicate with ROS modules.

Unlike previous solutions, the approach presented in this paper integrates the model definition with the model verification, execution, and monitoring using the RG framework. Most of the vehicle control frameworks are based on a middleware layer with several independent modules. These modules are connected with different inter-process communication mechanisms using the message publish/subscribe paradigm. This use of messages instead of events (interpreted PNs) or signals (signal interpreted PNs) establishes a significant difference with previous models.

The solution proposed here can be implemented in any control framework that includes a functional layer with a set of components that provide access to sensor data, send commands to actuators, and execute different basic functionalities (localization, path planning, obstacle avoidance, etc.) or behaviors (follow wall, etc.). However, the use of frameworks provides several advantages when the modules are implemented as independent processes, communicating with each other and with the upper layers via some inter-process communication mechanism. These include, for example, module reusability, robustness because a failure in a module does not imply the whole system failing, and easier maintenance.

Even though the executive layer presented here does not explicitly deal with the uncertainty about the state of the system, it allows the monitoring system to deal with it. The motivation of this approach is that most of the uncertainty comes from the observers that are directly handled by monitors. Therefore, once the decision about the current situation is taken, the sequence of actions to deal with it are known and, in our case, defined using PNs.

## Figures and Tables

**Figure 1 sensors-20-00449-f001:**
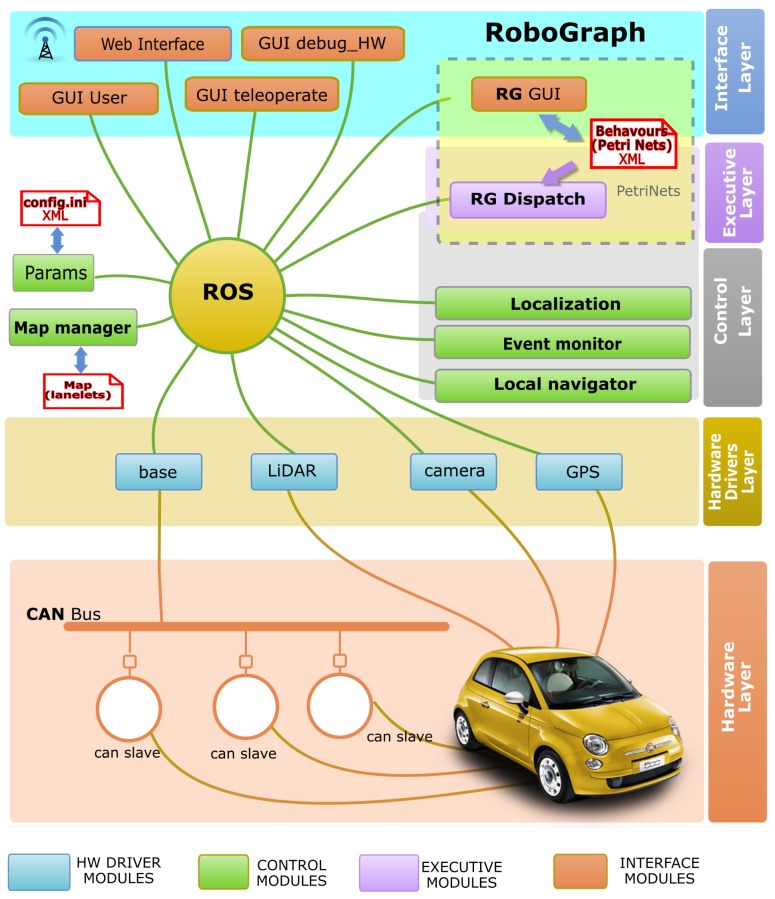
Car software architecture. ROS = Robot Operating System; GUI = graphical user interface; HW = Hardware; RG = RoboGraph; GPS = Global Positioning System; LIDAR = Light Detection and Ranging.

**Figure 2 sensors-20-00449-f002:**
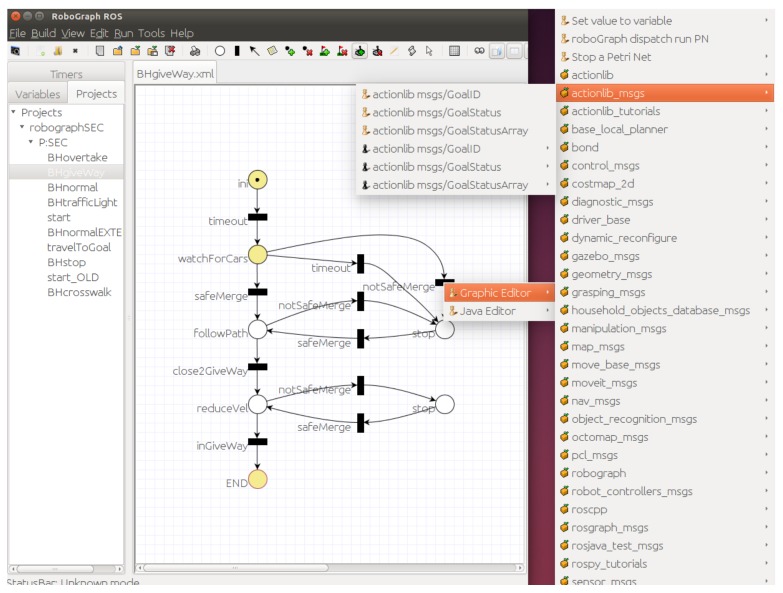
RG GUI editing the BHgiveWay Petri net (PN).

**Figure 3 sensors-20-00449-f003:**
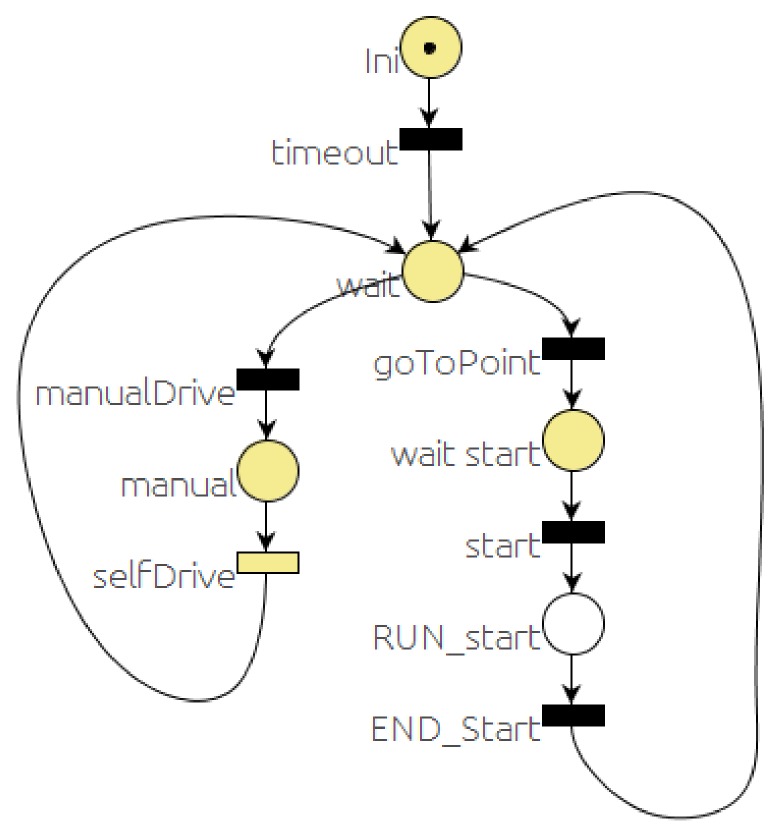
Latent PN.

**Figure 4 sensors-20-00449-f004:**
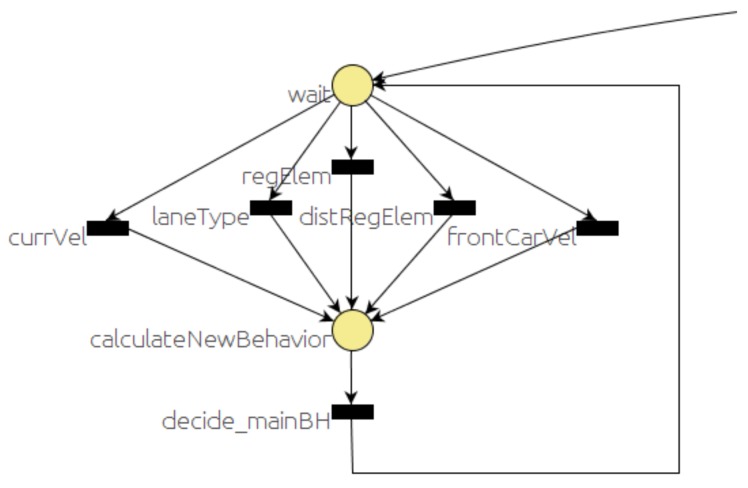
Selector PN: behavior decision part.

**Figure 5 sensors-20-00449-f005:**
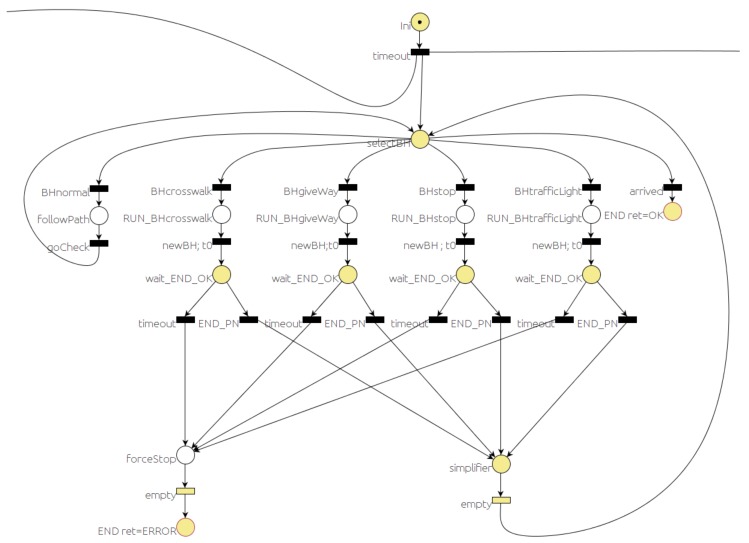
Selector PN: the behavior execution part.

**Figure 6 sensors-20-00449-f006:**
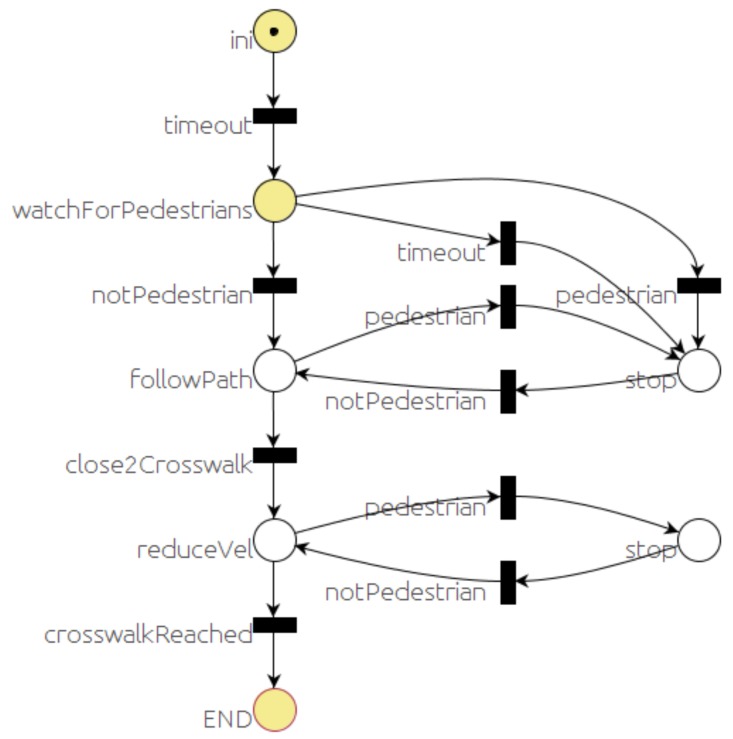
Crosswalk behavior PN.

**Figure 7 sensors-20-00449-f007:**
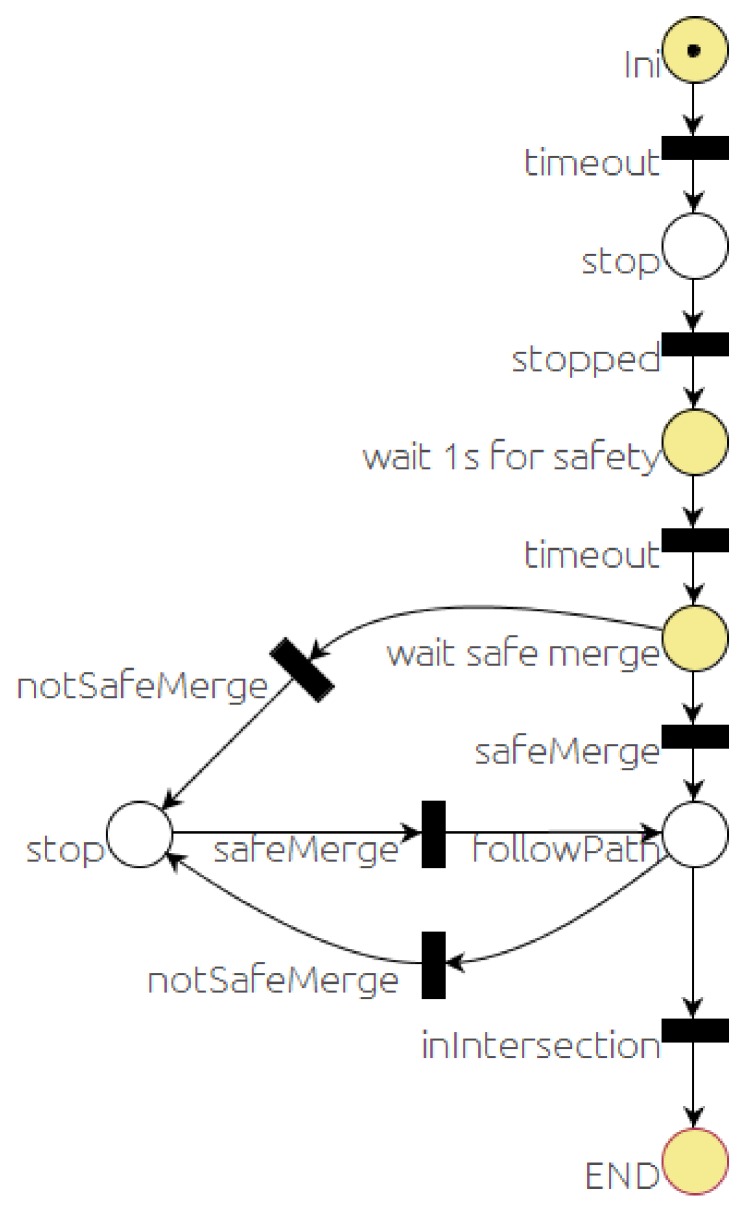
STOP behavior PN.

**Figure 8 sensors-20-00449-f008:**
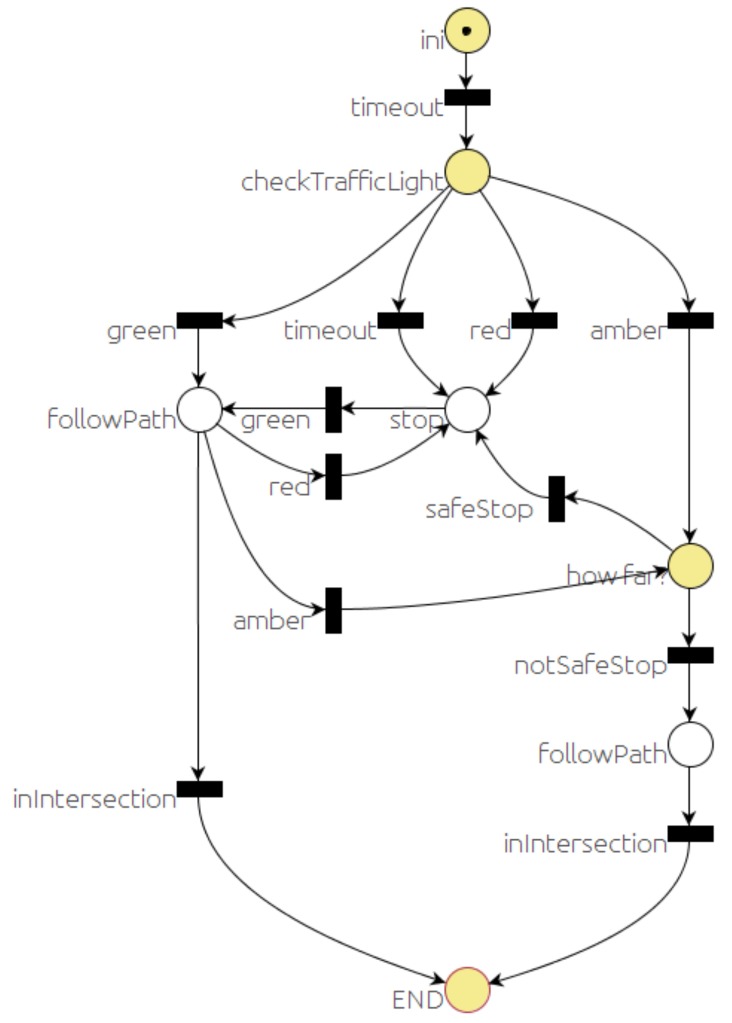
Traffic light behavior PN.

**Figure 9 sensors-20-00449-f009:**
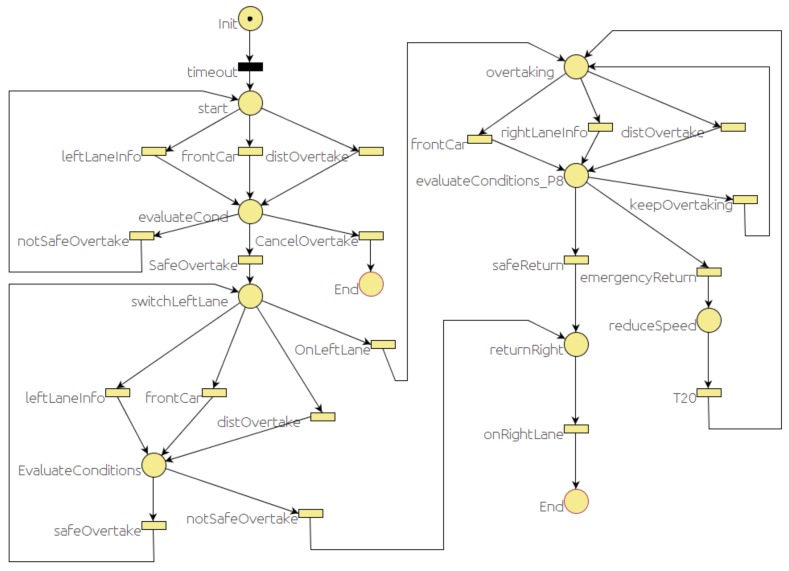
Overtake behavior PN.

**Figure 10 sensors-20-00449-f010:**
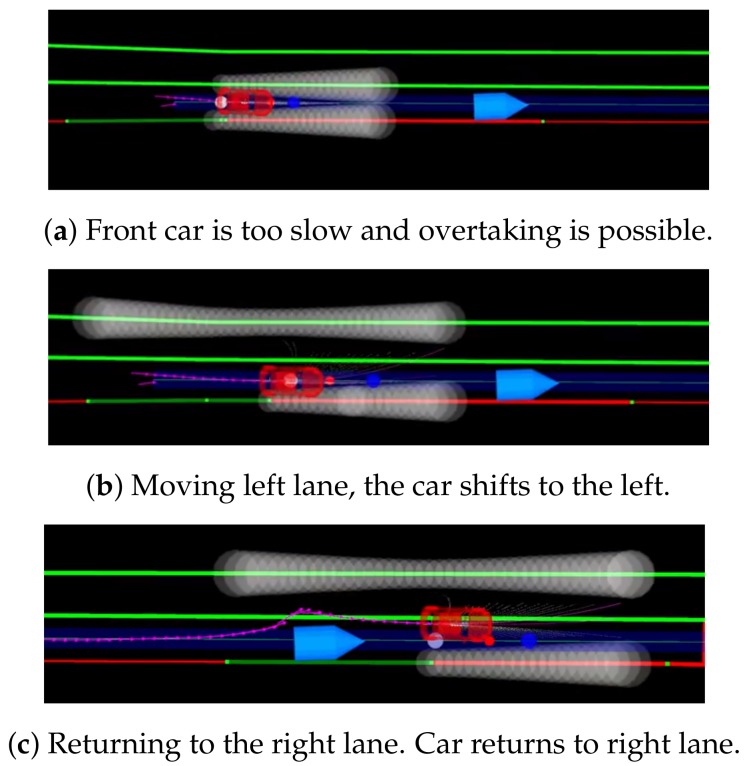
Overtaking maneuver.

**Figure 11 sensors-20-00449-f011:**
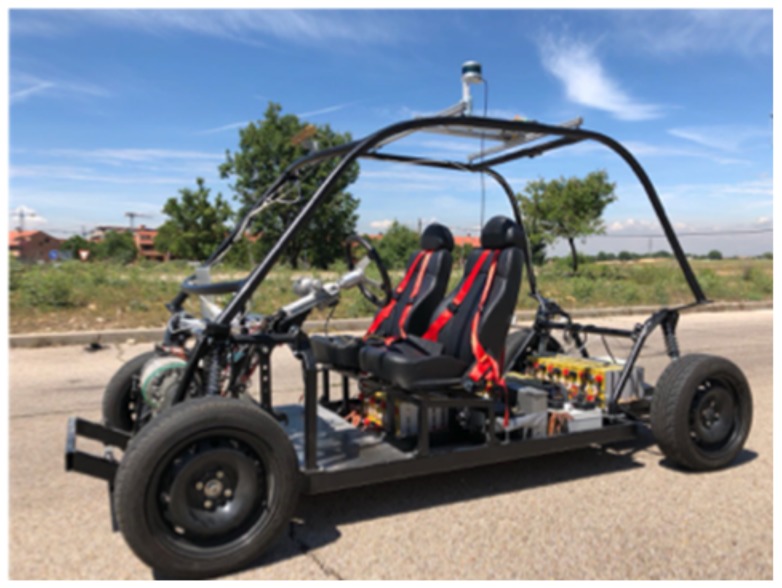
Car based on an Open Motors company TABBY EVO.

**Figure 12 sensors-20-00449-f012:**
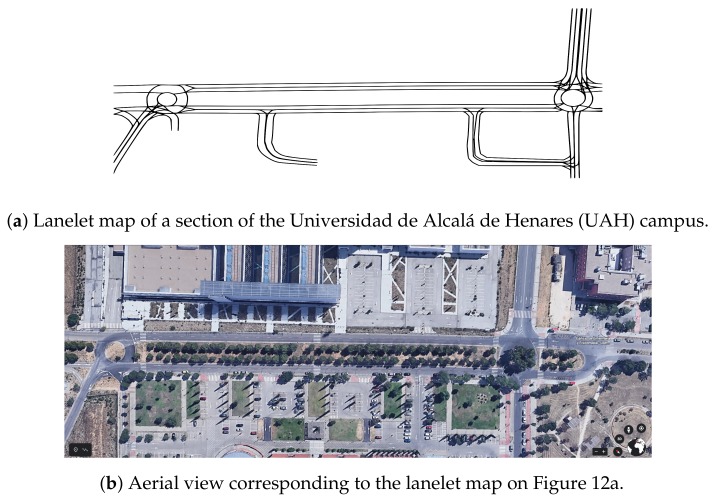
Overtaking maneuver.

**Figure 13 sensors-20-00449-f013:**
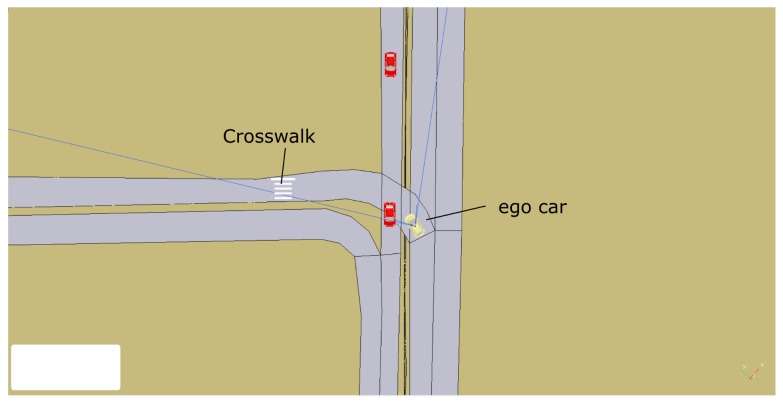
Stop scenario. Ego car in yellow has a STOP to turn to the left.

**Figure 14 sensors-20-00449-f014:**
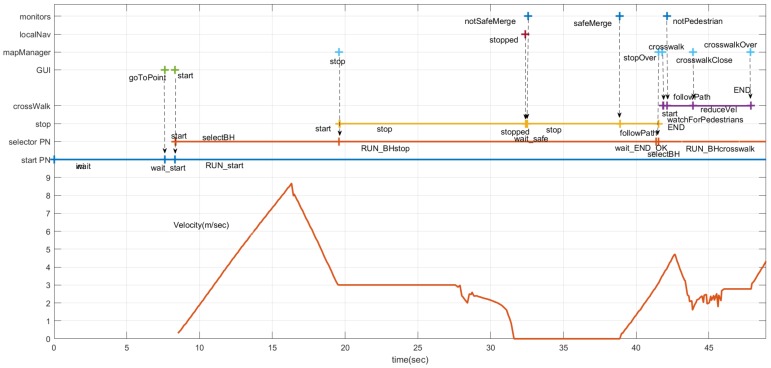
Temporal diagram. At the top, the events produced by modules monitors, localNav, mapManager, and GUI. In the middle, the evolution of crossWalk, stop, selector, and start PNs. At the bottom, the velocity commanded to the car.

**Figure 15 sensors-20-00449-f015:**
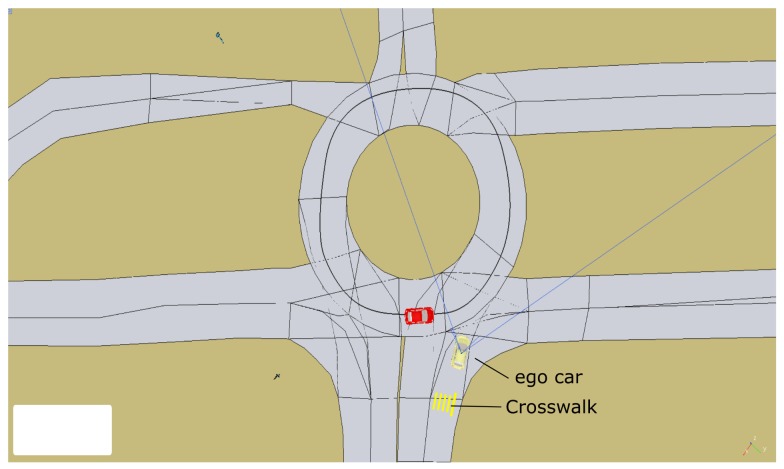
Entering a roundabout scenario. The ego car, in yellow, first encounters a crosswalk and then has to yield to other cars when entering the roundabout.

**Figure 16 sensors-20-00449-f016:**
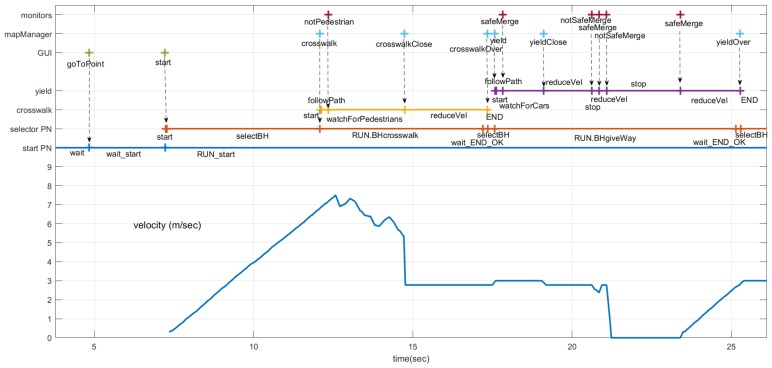
Temporal diagram for entering a roundabout scenario. At the top, the events produced by modules monitors, mapManager, and GUI. In the middle, the evolution of crossWalk, yield, selector, and start PNs. At the botton, the velocity commanded to the car.

**Figure 17 sensors-20-00449-f017:**
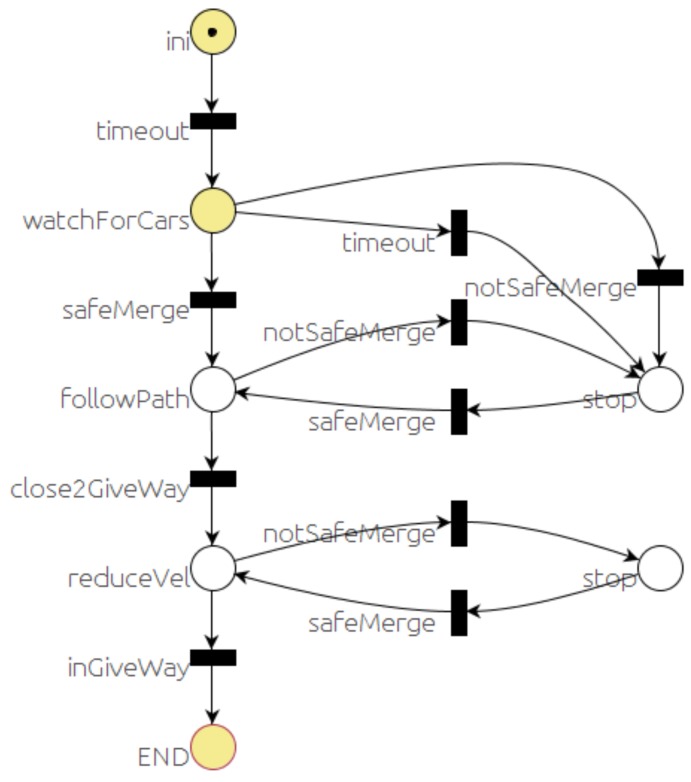
Yield behavior PN.

**Figure 18 sensors-20-00449-f018:**
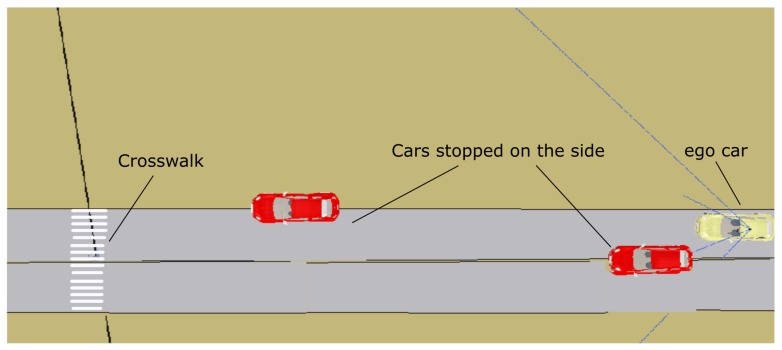
Avoiding obstacles and approaching a crosswalk. The ego car, in yellow, first encounters a couple of obstacles that partially obstruct the lane and then a crosswalk with a pedestrian using it.

**Figure 19 sensors-20-00449-f019:**
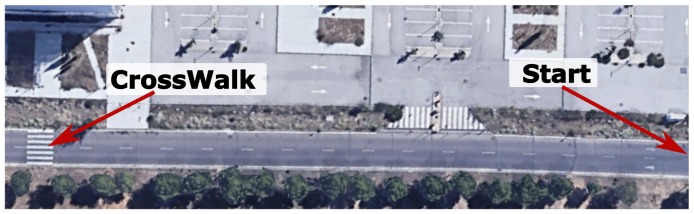
Section of the road of UAH map (Figure 12a) where the crosswalk with pedestrian scenario takes place.

**Figure 20 sensors-20-00449-f020:**
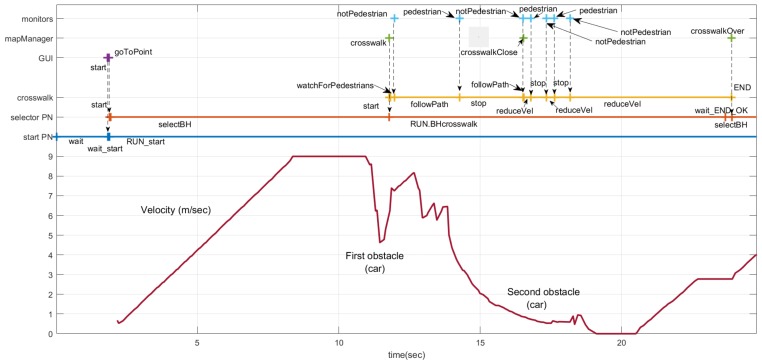
Temporal diagram for entering a roundabout scenario. At the top, the events produced by modules monitors, mapManager, and GUI. In the middle, the evolution of crossWalk, yield, selector, and start PNs. At the botton, the velocity commanded to the car.

**Figure 21 sensors-20-00449-f021:**
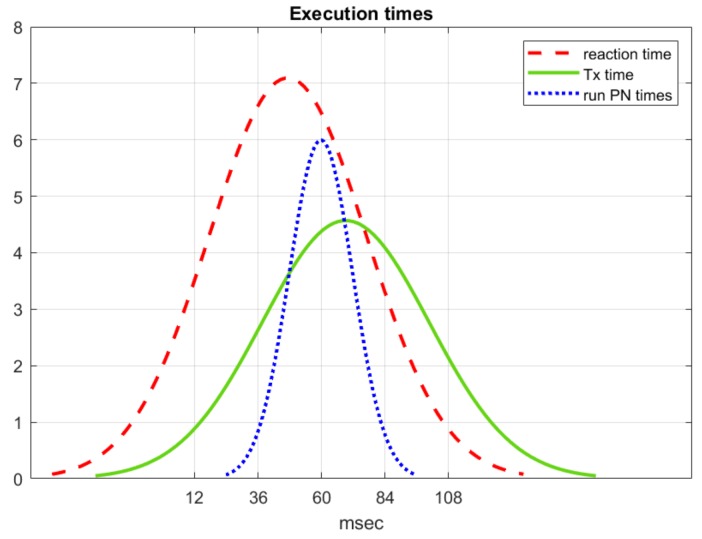
Reaction times of the executive layer.

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
