# Peer review of "Implementing Autonomous Driving Behaviors Using a Message Driven Petri Net Framework"

_sensors, 2020, doi:10.3390/s20020449_

Round 1

Reviewer 1 Report

In this paper, the authors investigated a message driven Petri net framework for implementing autonomous driving behaviors. This paper is well-written, with clear statements of the proposed methodology. However, The authors should make improvements to the manuscript based on the following perspectives:
1. There have been a lot of similar works on Petri net framework for autonomous driving. What are the novel findings in the present work?
2. The authors said"POMDP models cannot scale to complex real-world scenarios". Please provide more details on how to solve this problem of "complex real-world scenarios" by Petri net.
3. The authors sould provide more details on how to solve the"state explosion problem" and "uncertainty problem".

Author Response

Also, we would like to thank the reviewer for the careful reading, and constructive suggestions for our manuscript.

Reviewer 2 Report

1.English and punctuation need minor corrections which are annotated in the attached file.

2.At the end of the paper, the list of abbreviations is provided, but I believe that the meaning of several acronyms is missing: IPC, JIPC,BCM, CVM, IROS, OBJ. Please check that the meaning of each acronym is provided when mentioned for the first time in the main text, and is provided in the list of abbreviations at the end. 

3.Sometimes the authors write "PNs", sometimes "Petri Nets". I suggest the use of only one notation.

4.In figures 3 and 5, there are both black transitions and white transitions? What is the difference? Are they immediate transitions and timed transitions respectively? In figures 4, 6, 7, 8, 16, all the transitions are black; in figure 9 instead, all the transitions are white; is it correct?

5.At line 301 the authors say: "The first place and transition of all the Petri nets presented here include a timeout (300 msec)". What does "first place" mean? Is it the only marked place in the initial marking? Is the "first transition" the one enabled by the "first place"? Is the time-out modelled by a timed transition different from the other ones? 

6.In the abstract the authors mention "hierarchical interpreted binary Petri nets". What does "binary" mean? Does it mean that a place can contain 0 or 1 token? Is there a specific replication/join model to construct the composed Petri Net from the basic Petri Nets?

7.There is a dangling arc in figure 4, and there is there is a dangling arc in figure 5. Are they the same arc connecting the two Petri Nets? Figure 5 contains a second dangling arc; what is its destination?

8.In figure 10, image (c) overlaps the caption of image (a). The second line of the caption of image (b) seems to be the continuation of the caption of image (a); more space would be necessary between the captions of images (a) and (b).

9.At the end of the paper, the section containing conclusions and future work is missing. At line 153, the authors talk about "subsystems running in parallel"; so I believe that a possible future work could be the use of Coloured Petri Nets.

10.The introduction should be section 1.

Author Response

(The authors gave the same response as above.)

Reviewer 3 Report

Dear authors,

        you put forward a novel method to implement autonomous driving behaviors using a discrete even model framework to assist the elderly people with driving tasks.

Overall, the paper is well written. The descriptions are clear and precise. Experiments/field tests are well designed and results are well demonstrated. Therefore, I do not have any major concerns about what has been presented in the paper.

        I have some minor suggestions and questions:

        1. In Figure 1, IPC should be spelled out. Why is JIPC not included in figure 1?

        2. In Figure 1, Velodyne should be LiDAR to be more general. In addition to camera and lidar sensors, other perception modules like image segmentation used to detect the basic events, are suggested to be included in Figure 1.

        3. Figure 10 is in a mess, which should be fixed.

        4. In Figure 11, the smart elderly car is similar to the fully electric vehicles in [R1][R2][R3][R4][R5], which should be mentioned. Different components/modules of the car should be introduced (in Figure 11 or in the text). As it is claimed, the car is designed for elderly people, but which part of the system is specifically related to elderly people's needs?

        5. For the road tests (section 6.1), do you have some photos of the UAH campus or the IROS event, to present the scenarios, along with Figure 12 and Figure 14?

        6.  In crosswalks with pedestrians, how do you detect the obstacles/pedestrians? 

        7. What do you want to demonstrate in Figure 19? Please consider deepening in the analysis.

        8. Important references missing:

[R1] Romera, E., Bergasa, L.M., Yang, K., Alvarez, J.M. and Barea, R., 2019, June. Bridging the day and night domain gap for semantic segmentation. In 2019 IEEE Intelligent Vehicles Symposium (IV) (pp. 1312-1318). IEEE.

[R2] Yang, K., Hu, X., Chen, H., Xiang, K., Wang, K. and Stiefelhagen, R., 2019. DS-PASS: Detail-Sensitive Panoramic Annular Semantic Segmentation through SwaftNet for Surrounding Sensing. arXiv preprint arXiv:1909.07721.

[R3] Tradacete, M., Sáez, Á., Arango, J.F., Huélamo, C.G., Revenga, P., Barea, R., López-Guillén, E. and Bergasa, L.M., 2018, November. Positioning system for an electric autonomous vehicle based on the fusion of multi-gnss rtk and odometry by using an extented kalman filter. In Workshop of Physical Agents (pp. 16-30). Springer, Cham.

[R4] Sáez, Á., Bergasa, L.M., López-Guillén, E., Romera, E., Tradacete, M., Gómez-Huélamo, C. and del Egido, J., 2019. Real-Time Semantic Segmentation for Fisheye Urban Driving Images Based on ERFNet. Sensors, 19(3), p.503.

[R5] Yang, K., Hu, X., Bergasa, L.M., Romera, E. and Wang, K., 2019. PASS: Panoramic Annular Semantic Segmentation. IEEE Transactions on Intelligent Transportation Systems.

[R6] Son, S., Jeong, Y. and Lee, B., 2019. An audification and visualization system (AVS) of an autonomous vehicle for blind and deaf people based on deep learning. Sensors, 19(22), p.5035.

[R7] Harper, C.D., Hendrickson, C.T., Mangones, S. and Samaras, C., 2016. Estimating potential increases in travel with autonomous vehicles for the non-driving, elderly and people with travel-restrictive medical conditions. Transportation research part C: emerging technologies, 72, pp.1-9.

For these reasons, revision is suggested.

Yours sincerely, 

Author Response

(The authors gave the same response as above.)

Reviewer 4 Report

The authors present a framework based on Petri Nets to design an autonomous driving agent via a graphical interface called RoboGraph GUI.

In the following, I present some main questions related to the paper.

Is it not clear if the authors participate in the development of the RoboGraph GUI, if yes clarify this point in the introduction or in the abstract, if not please there are not reasons to add one entire section to the RoboGraph tool.

In section 1 no space is given to autonomous vehicle architecture based on robotic subsumption architecture, e.g. some inspired from the neurocognition of human driving.

In this paper, a Petri net that implements an autonomous agent is presented. Different behaviors are used to deal with different situations. In section 4.2 line 323 the authors write that when the traffic situation change the token from "wait" state is removed and a new token is placed in "calculateNewBehaviour" state. What happens to current behavior, if the maneuver is already started and the agent starts a new maneuver? If the situation goes back as before the agent can restore the previous state?

Figure 5 seems that the system looking for pedestrian-only in the crosswalk. What happens if a pedestrian crosses out to the crosswalk? Can the system manage an overtake to avoid a pedestrian? How different behaviors interact together? 

A conclusion section must be added.

Minor changes or questions:

The acronym IPC is missing. The acronym BCM is missing. Please introduce the RG dispatch module in the section 3.2. In the section 4.1, the authors refer to figure 3 as a simplified version of the Latent Petri net. Why is simplified? Why is not presented the one used in the simulation? In section 4.1 line 309 the transaction from "wait" is "RUN_start" but from the figure 3 is resulting "wait start". In figure 4 from where the arrow that goes to "wait" state comes? Does it come from "ini" figure 5?. The section 5.2 is called Intersection but figure 7 is called STOP and figure 8 is traffic light, it is not clear. These two behaviors can run in parallel? Figure 10 must be revised, the captions are under the figure.  Figure 13 shows the car reducing the speed to 3m/s before turning left, Why this behavior? 

Author Response

(The authors gave the same response as above.)

Round 2

Reviewer 1 Report

I suggest this manuscript can be published. However, some more detailed description of the real road test in the result section would improve the paper.

Author Response

We would like to thank the reviewer for your contributions to improve the manuscript.

Reviewer 2 Report

The paper needs minor corrections:

line 135: FSM ---> (FSM)
line 172: The architecture focus on sensing and reaction and builds ---> The architecture focuses on sensing and reaction, and builds
line 255: between them ---> between modes
line 266: process ---> processes
line 270: open the log file and play ---> open the log file, play

The authors talk about the "background colour" of places and transitions.
In my opinion, it is the foreground colour.

What is the meaning of the acronym DARPA?
What is the meaning of the acronym UAH? I believe that it means "University of Alcalá de Henares", but it is not clearly expressed.

Author Response

(The authors gave the same response as above.)

Reviewer 4 Report

I appreciated all the changes provided by the authors.

My only concern is on the abstract that is defective in explaining what is the major contribution of the authors (as now explained in the conclusion). This is my suggestion to add some information in the abstract:
... Here we propose an executive module (... Here I would add something that clarifies that a graphic interface for develop and debug is introduced and adapted for ROS and is called Robograph) ...

Author Response

(The authors gave the same response as above.)
